# Dietary Behaviors and Metabolic Syndrome in Schizophrenia Patients

**DOI:** 10.3390/jcm9020537

**Published:** 2020-02-16

**Authors:** Katarzyna Adamowicz, Jolanta Kucharska-Mazur

**Affiliations:** Department and Clinic of Psychiatry, Pomeranian Medical University in Szczecin, 26 Broniewskiego Street, 71-460 Szczecin, Poland; jola_kucharska@tlen.pl

**Keywords:** schizophrenia, metabolic syndrome, eating behaviors, cognitive performance, self-awareness

## Abstract

The metabolic syndrome (MS) is highly prevalent in schizophrenia patients, resulting from both pharmacotherapy and their lifestyle. To avoid its development, the analysis of patients’ eating behaviors followed by the necessary nutritional changes should become a routine element of treatment. The aim of this study is to investigate the effect of dietary habits on the course of schizophrenia and MS, cognitive performance, symptom severity, and subjective assessment of eating behaviors in schizophrenia patients. Total of 87 participants (63.2% women) aged 19 to 67 years (M = 41.67; SD = 12.87), of whom 60 met the IDF criteria for MS, completed the PANSS, the verbal fluency test, the Stroop Color-Word Test, and the digit span task, followed by a thorough nutritional interview. There were no significant differences in the dietary behaviors between investigated schizophrenia patients with and without comorbid MS. Interestingly, their eating habits compared quite favorably to those described in the literature. No associations were found between positive eating habits and other tested variables in patients with MS. They were, however, linked to lower PANSS scores in the entire sample. In addition, positive eating habits correlated with better cognitive performance and a more adequate subjective assessment of dietary habits. It would be amiss to assume that schizophrenia patients lack the ability to control their eating behaviors. Nutrition education may foster desirable dietary changes and improve the sense of agency, thus helping to reduce symptom severity and enhancing cognitive performance in this patient population.

## 1. Introduction

Because of pharmacotherapy and lifestyle choices, persons diagnosed with schizophrenia are at a higher risk of developing the metabolic syndrome [1,2,3,4]. Given the importance of dietary regimen in the aetiology of various medical conditions [5,6,7], and the clear links between mental disorders and the incidence of metabolic diseases, the relative paucity of scientific papers on the effect of nutrition on mental health remains quite surprising. Modification of dietary habits is one of the available non-pharmacological interventions recommended to psychiatric patients to improve their general health. Although there is abundant evidence suggesting considerable effect of neuroleptics on obesity and carbohydrate metabolism disorders, weight gain in patients with schizophrenia is also observed prior to the introduction of pharmacotherapy [8]. Moreover, not all patients who receive antipsychotic treatment put on weight at the same rate, suggesting interindividual variations in the magnitude of weight increase. Therefore, regardless of the adverse effects of pharmacotherapy, it is the lifestyle choices that seem to be of key importance in the development of metabolic concomitants of mental disorders [9].

Schizophrenia patients are known to make poor dietary choices and have an excessive daily food energy intake. Their menus include large amounts of animal fats, while their intake of vegetable fats is clearly lower compared to healthy population [10]. Large supply of saturated fats alongside low fiber and fruit intake are associated with elevated levels of inflammatory markers, especially tumor necrosis factor (TNF), interleukin 6 (IL-6), and C-reactive protein (CRP), which may promote the development of MS [11]. According to the reports of the World Health Organization (WHO), countries with a larger total intake of dietary fat and saturated fat from land animals have a higher prevalence of schizophrenia than those reporting a relatively higher intake of polyunsaturated fatty acids found in fish or seafood [12]. Data from a group of 194 schizophrenia patients showed that they did not follow a regular eating schedule, were likely to have their meals mainly in the morning and evening, drank more coffee and consumed more instant meals compared to healthy controls. They also smoked more than the control group (on average four cigarettes more per day). Of note, insufficient physical activity and poor dietary, drinking, and smoking habits proved to be particularly adversely affected by schizophrenia in connection with unemployment [13]. Another study demonstrated that 51% of the 159 included schizophrenia patients consumed their meals in less than 15 min, 41% did not eat a single fruit per day, and 63% did not eat fish [14]. Various nutrient deficiencies have been observed in people with schizophrenia, as they were found to meet less than 75% of their daily fiber, magnesium, thiamine, folic acid, and vitamin D requirements [15]. In a Turkish study, a group of 62 female and 42 male outpatients with schizophrenia were assessed for food addiction with the Yale Food Addiction Scale. The study found that over half of the participants (60.6%) manifested food addiction, with a higher prevalence in women (62.9%) than in men (57.1%). As expected, the food addicts had higher total energy intake and consumed significantly more carbohydrates and fats [16].

Visceral obesity occurs in 40–60% of schizophrenia patients, and the accumulation of large amounts of abdominal fat has considerable health consequences. Long-term schizophrenia treatment is associated with approx. 2.5-fold greater risk of developing MS in female patients and approx. 1.5-fold greater risk in male patients compared to the general population [17]. A study on a group of 159 patients with schizophrenia found comorbid obesity in as many as 83.6% of the sample [14]. As the main component of MS, abdominal obesity significantly deteriorates daily functioning of patients and entails various psychological problems. Pharmacotherapy-induced self-regulation deficits, resulting in increased appetite, but also feelings of helplessness, may lead to an excessive food intake, and consequently obesity. Concomitant obesity may be associated with reduced self-esteem and negative self-perception. People struggling with excess weight or obesity are often perceived as poorly motivated, self-neglecting, lacking self-control, and generally unambitious [18].

There are reports suggesting that the presence of MS or its components may be associated with reduced cognitive performance. Obesity alone carries a greater risk of developing neuropsychiatric disorders. Patients with schizophrenia and concomitant MS perform worse in memory, executive functions, and attention compared to those without MS [19,20]. Reduced cognitive performance is also believed to affect motivation. Negative symptoms of schizophrenia may contribute to motivational deficits, resulting in decreased involvement in various activities and reduced goal-oriented behaviors. The sense of agency, consisting of subjective control over the course of events, a sense of self-efficacy and freedom of choice, are suggested to be impaired in schizophrenia [21,22]. Therefore, it is often assumed that patients may lack the capacity to persevere, e.g., in their endeavors to alter their dietary habits. On the other hand, it has been shown that the use of interventions enhancing intrinsic motivation can improve attention and learning in schizophrenia patients, resulting in better management of their disease and increased perceived competency [23].

The following research hypotheses were formulated to investigate the effect of dietary habits on the metabolic syndrome and cognitive functioning in patients suffering from schizophrenia:Patients with schizophrenia and concomitant metabolic syndrome manifest poorer eating habits compared to those without the metabolic syndrome.Poor eating habits are associated with poor cognitive performance in patients with schizophrenia.Poor eating habits are associated with higher positive and negative PANSS scores.Schizophrenia patients’ subjective assessment of their dietary behaviors is more favorable compared to an objective dietary assessment in this patient population.

Presented research results are part of a larger project on eating habits, metabolic syndrome, and dietary interventions in patients with schizophrenia. The study was approved by the Bioethical Committee of the Pomeranian Medical University in Szczecin (Resolution No. KB-0012/108/14).

## 2. Material and Method

### 2.1. Statistical Analysis

Statistical analysis was performed with the IBM SPSS Statistics v.25 package, using non-parametric comparison (the chi-square test and the Mann–Whitney U test) and correlation methods (Spearman’s rho). Statistical significance was set at *p* < 0.05, while the *p* value of <0.1 was considered a statistical trend.

### 2.2. The Sample

The study included 87 patients with schizophrenia (F20 according to ICD-10), aged 19 to 67 years. Detailed sociodemographic characteristics of the sample are presented in Table 1. Having provided a written consent to participate in the study, each patient underwent a mental state examination by a licensed psychiatrist to verify the diagnosis of schizophrenia and remission status (using the PANSS scale), followed by a physical examination, anthropometric measurements, and laboratory tests. The metabolic syndrome was diagnosed according to the International Diabetes Federation (IDF) criteria. Exclusion criteria were other serious medical conditions that could require specialized dietary regimen or significantly affect cognitive performance. Such as acute symptoms of schizophrenia (increases positive and negative symptoms), eating disorders, known food intolerances, dementia, addiction to psychoactive substances, kidney and liver problems, thyroid dysfunction, syndromes of cardiac function disorders, oncological diseases. Administered pharmacotherapy included olanzapine, clozapine, quetiapine, and aripiprazole.

### 2.3. Research Tools

Symptom severity was assessed with the PANSS (Positive and Negative Syndrome Scale), translated by Małgorzata Rzewuska [24]. This 30-item tool enables assessment of positive (P), negative (N), and general psychopathology (G) symptoms on a 7-point scale (1—absent, 2—minimal, 3—mild, 4—moderate, 5—moderately severe, 6—severe, 7—extreme) [25].

Metabolic syndrome is associated with the coexistence of various risk factors that can cause cardiovascular disease and type 2 diabetes. There are no commonly accepted criteria for metabolic syndrome, but it is generally acknowledged to include abdominal obesity, glucose intolerance, hypertension, and dyslipidemia. To diagnose the metabolic syndrome in accordance with the IDF criteria (*International Diabetes Federation)* of 2005, anthropometric measurements, blood pressure measurement, and laboratory tests were performed. A prerequisite was the occurrence of abdominal obesity (i.e., waist circumference of >94 cm in men and >80 cm in women), and any two of the following: raised triglycerides (>150 mg/dl); reduced HDL cholesterol (<40 mg/dl in men, <50 mg/dl in women); raised blood pressure (>130/85 mmHg); raised fasting plasma glucose (>100 mg/dl) [26]. The anthropometric measurements included height, weight, hip, and waist circumference. Height was measured using a height meter with an accuracy of ±0.1 cm, in an upright position, without footwear. An electronic balance was used to measure the body weight with an accuracy of ±0.1 kg. Blood levels of total cholesterol, LDL, HDL, triglycerides, and glucose were determined in all patients. All test results were compared to reference values in accordance with the recommendations of the Polish Cardiac Society and the Polish Society of Laboratory Diagnostics.

Cognitive performance assessment was based on:

The Stroop Color-Word Test (SCWT)—used to measure processing speed, verbal working memory, and executive functions related to habitual response inhibition and the ability to switch to a new response criterion [27,28],

The verbal fluency test (VFT)—to measure semantic and phonemic verbal fluency [29],

The Backward Digit Span Task—used to measure the ability to store and manipulate objects in memory, related to the functioning of verbal working memory [30].

A standardized medical history (including nutritional history) provided sociodemographic data and information on patients’ eating habits. The first part of the interview included questions on: sex, age, education, place of residence, marital status, occupational status, comorbid disorders, disease duration, number of hospitalizations, patient’s knowledge on the metabolic syndrome, administered pharmacotherapy, family history of mental and metabolic diseases.

The second part of the interview covered diet-related factors and contained questions about:Diet (data concerning food intake in the last 24 h and assessment of the most typical meals in the patient’s menu);Places where meals are consumed;Mealtimes;Applied culinary techniques;Dietary supplements;Weight loss diets;Daily consumption of fruit and vegetables;Daily consumption of cereal products;Daily consumption of dairy products;Daily consumption of meat products;Daily consumption of fats;Weekly consumption of fish;Daily consumption of sweets and salty snacks;Daily consumption of carbonated soft drinks, fruit juices;Use of cigarettes and alcohol;Use of salt and other condiments;Subjective assessment of dietary habits—rated on a 5-point scale (5—very good—the patient is satisfied with his/her eating habits, claims that his/her diet does not require any changes and is consistent with the principles of healthy eating; 4—good—the patient claims to be healthy; however, certain deviations from the principles of healthy eating sometimes occur and he/she could introduce some changes; 3—sufficient—the patient pays attention to products he/she consumes, but is inconsistent; aware of the mistakes he/she makes, claims that “it could still be worse”; 2—poor—the patient does not eat healthily, assesses his/her diet as poor and notes that it would make sense to change their eating habits; 1—insufficient—the patient does not pay attention to what he/she eats, caves to momentary cravings, is fully aware of the fact that their diet is very unhealthy).Physical activity—rated on a 3-point scale (1—low physical activity, sedentary lifestyle; 2—moderate physical activity, no conscious decision to include exercise into daily activity, physical activity resulting from daily duties; 3—high physical activity, conscious and frequent exercise, increased activity).

The above dietary behaviors and matching variables were divided into two groups, following nutrition recommendations for the Polish population based on the “Pyramid of Healthy Nutrition and Physical Activity” amended in 2016 and the updated “Polish Dietary Reference Intakes,” developed under the guidance of prof. Mirosław Jarosz at the Institute of Food and Nutrition in Warsaw [31]:Positive eating habits (drinking water, proper intake of fruit and vegetables, whole grain products, fish, preference for baking and stewing, regular dietary regimen, systematic use of pharmacotherapy, regular physical activity).Negative eating habits (use of alcohol, drugs; smoking; excessive use of salt; consumption of fast foods, sweets, large amounts of meat, snacking between main meals; irregular diet; preference for fried meals; excessive intake of coffee, sweetened fruit juices, poor compliance to pharmacotherapy, low physical activity).

## 3. Results

Nutritional assessment showed schizophrenia patients’ eating habits and cooking preferences, as well as diet-related differences between patients with and without comorbid MS. Because of the nominal nature of data, differences in consumption patterns of selected food products were tested with chi-square statistics.

The vast majority of our sample reported frequent consumption of healthy products such as dairy, fruit, and vegetables (over 90% of the entire sample), although consumption of sweets was also very common (almost 89%). The patients preferred water (88%) over juices (59%) or carbonated soft drinks (46%). The least popular products were fast foods and salty snacks (consumed by 39% and 41% of patients, respectively). Total of 46% of respondents were smokers, about 36% reported drinking strong coffee, and 21% strong tea. Only >6% of the sample admitted to drinking alcohol. There were no significant differences in consumed food products or stimulants between patients with and without MS. A detailed description of the dietary habits reported by the investigated schizophrenia patients is presented below, in Table 2.

The last part of Table 2 provides information on the preferred cooking methods in the sample. The most commonly reported culinary technique was frying (87%), followed by boiling (64%). Every third patient preferred baking, and fewer than 1/5 of the sample indicated simmering as their favorite cooking method. Two significant differences were observed between patients with and without MS; frying was clearly more popular among patients with MS (95% vs. 70% in the control group; *p* < 0.01), while patients without MS tended to mainly boil their meals (81.5% vs. 57% in MS patients; *p* < 0.05).

As shown in Table 3, the two groups did not differ in terms of the number of positive and negative eating habits, subjective assessment of dietary habits, symptom severity, or cognitive performance. The only correlation observed in the entire sample was the one between positive symptom severity and positive eating habits (rho = −0,220; *p* = 0,041), indicating that less severe positive symptoms were associated with healthier eating behaviors. However, a number of relationships emerged after dividing the sample according to the occurrence of comorbid MS (Table 4 and Table 5).

In patients with MS, positive eating habits were not associated with any of the other analyzed variables, while there were weak negative correlations between negative eating habits and negative symptom severity (rho = −0.305; *p* = 0.018). In addition, weak positive links were observed between subjective assessment of dietary habits and positive symptom severity (rho = 0.277; *p* = 0.032).

We found a number of associations between eating habits, schizophrenia symptom severity, and cognitive performance in patients without comorbid MS. There was a moderate negative correlation between positive eating habits and the severity of positive (rho = −0.501; *p* = 0.008) and general symptoms (rho = −0.515; *p* = 0.006), and a weaker, albeit borderline significant, correlation with the negative symptoms (rho = −0.369; *p* = 0.059). In addition, healthier eating habits were linked with better performance on the Stroop test (rho = −0.616; *p* = 0.001). In this group, none of the participants made a single error, hence the observed lack of correlations for this variable. We observed a statistical trend indicating improved verbal fluency (the “animals” category) linked to healthier eating habits (rho = 0.383; *p* = 0.054) and poorer performance on the Forward Digit Span Task (rho = −0.354; *p* = 0.076).

Poorer eating behaviors were associated with subjective assessment of dietary habits (rho = −0.485; *p* = 0.010) and a tendency to perform poorer on the Forward Digit Span Task (rho = −0.354; *p* = 0.076)—neither of which were observed among patients with MS.

We also checked if the groups differed in terms of administered pharmacotherapy. The only difference was found in the use of aripiprazole, administered to 48.1% (*n* = 13) patients with MS and 13.3% (*n* = 8) patients without MS (X(1) = 12.325; *p* = 0.001). Nevertheless, aripiprazole treatment did not seem to affect either the positive (Z = −0.878; *p* = 0.380) or the negative eating habits (Z = −0.123; *p* = 0.902) reported by the entire sample, or patients with and without MS analyzed separately (*p >* 0.05).

## 4. Discussion

Somewhat contrary to the stereotypical perception of a psychiatric patient as a person with little knowledge and little awareness of the surrounding phenomena, the study sample included mostly patients with secondary and higher education. It can therefore be assumed that it is the lack of nutrition education rather than lacking intellectual resources or poor general knowledge that affects the quality of nutrition. Both groups were predominantly composed of big city residents who received disability benefits because of mental illness. However, the sample also included patients who, despite illness-related functional deficits, pursued employment. However, unemployment translates into more free time, which can be used to introduce changes into one’s eating habits, especially when pharmacological treatment is associated with metabolic side effects. Nevertheless, given the aforementioned reports by Roick et al. suggesting that the coincidence of schizophrenia and unemployment can contribute to poor eating habits and low physical activity [13], it is paramount for patients to be properly supported by specialists in their efforts to use their leisure time effectively. Patients who remain employed are a living proof that mental illness does not necessarily preclude active functioning and responsible decision-making. Their sustained professional activity thus remains in contrast with the popular belief (also supported by some research) that schizophrenia patients have low economic status, suffer from amotivation, and manifest functional apathy [32,33].

In our sample, there were only few significant diet-related differences between patients with and without comorbid MS. Regardless of the concomitant diagnosis of MS, they reported frequent consumption of dairy products, fruits, vegetables, even fish, and only few (less than half of the sample) admitted to eating fast foods, which contrasts with the reports by Simonelli-Munoz et al. [14]. Excessive body weight and the severity of metabolic symptoms in the study participants could be accounted for by their frequent intake of sweets. Ratliff et al. investigated the metabolic profile in relation to eating habits and physical activity in individuals with schizophrenia. According to their results, they manifested higher levels of glycosylated hemoglobin and insulin compared to matched controls. Their daily energy intake was no different from that of healthy individuals, but they consumed significantly greater amounts of fat and sugar. Schizophrenia patients also reported significantly lower physical activity [33]. Anderson et al. demonstrated the key importance of the quality of nutrition. Snacks in the form of desserts or sweets was associated with higher daily calorie intake (2076 kcal) than in the control group (1703 kcal) [34]. Strassing et al. found that the dietary choices made by schizophrenia patients did not necessarily differ from those reported in the general population, the reason for their excessive body weight being rather the effect of excess energy balance, resulting from significantly larger daily food intake [35]. Recent research shows that population obesity is primarily associated with an excess caloric quantity of the food supply, and not with the consumption of any specific individual dietary macronutrient [36]. Consumption of sweet snacks may be linked to a significantly higher intake of fat, saturated fatty acids, and lower intake of protein or fiber. Consumed sugar may be associated with lower survival rates. In a cohort of older adults who consumed relatively large amounts of sugar over 10 years, Anderson et al. recorded a relatively higher (by 37%) risk of mortality [34].

In this study, both patient groups significantly differed in terms of the applied culinary techniques. The vast majority of patients with concurrent MS reported deep fat frying as their preferred form of cooking, which may be associated with elevated inflammation markers [11]. Studies suggest that patients with schizophrenia often have a very monotonous, undifferentiated diet and do not pay attention to the preparation of consumed products [10]. Our results also demonstrate that excessive consumption of sweet snacks and the use of deep-frying techniques in the preparation of meals are associated with their significantly higher caloric content, often exceeding daily requirements thereof. Therefore, it seems that a positive energy balance may well be one of the key obesity-related factors in people with schizophrenia, and one that is likely to significantly contribute to future development of the metabolic syndrome. Contrary to patients with concomitant MS, those without MS mostly preferred culinary techniques such as boiling, simmering, or baking.

Of all the metabolic changes in schizophrenia, obesity and dyslipidemia seem to manifest most adverse effects on cognitive performance. Excessive body weight in the early stages of psychosis may constitute an independent risk factor for diffuse brain changes [20,37]. We observed an association between unhealthy eating habits and negative symptom severity in patients with MS. Links between cognitive performance and the severity of schizophrenia symptoms were demonstrated by Zhang et al. who reported greater negative and general symptom severity associated with reduced cognitive function in schizophrenia patients [38]. Data show that patients with severe negative symptoms manifest major deficits within the functioning of working memory [39]. Therefore, it could be postulated that over time, if the quality of their nutrition does not change and the negative symptoms persist, persons with MS are at a higher risk of developing cognitive impairment. Neuropsychological assessments of 216 schizophrenia patients demonstrated that those with MS had lower immediate memory and attention scores than those without MS [38]. In another study, patients with schizophrenia and MS (*n* = 159) performed poorer in tests measuring processing speed, attention, working memory, reasoning and problem solving compared to the group without MS [40].

In our sample, positive eating habits were associated with better cognitive functioning. Because of the correlational research design, it remains unclear whether positive eating habits increase cognitive performance, or better cognitive function results in healthier dietary choices. Such potentially bidirectional associations were also observed between dietary habits and positive symptoms. If greater positive symptom severity and poor cognitive function result in unhealthy dietary patterns, then we should strive for the most effective treatment of these symptoms. The inverse relationship (i.e., healthy diet reduces the severity of positive symptoms and improves cognitive performance) would have equally serious consequences, making education and dietary interventions a valuable adjunctive treatment. There is evidence that dietary habits consistent with the principles of rational nutrition could improve cognitive performance. Research reports reveal protective effects on the central nervous system of the B group, C and E vitamins, folic acid, and omega-3 fatty acids [41,42,43,44]. The human brain is composed of approximately 20% of omega-3 and omega-6 fatty acids, which must be supplied from food [45]. Clinical trials suggest that brain-derived neurotrophic factor (BDNF) is one of the biomarkers that are related to cognitive functioning. There are significant relationships between MS and reduced BDNF levels, affecting synaptic plasticity. A low energy diet can contribute to higher BDNF levels [38,46]. Positive eating habits, i.e., avoiding fast-food meals, sweets or carbonated beverages other than mineral water could significantly reduce daily calorie intake.

An important aspect of altering dietary patterns in schizophrenia patients is their subjective approach toward their eating habits. Links between positive symptoms and distorted assessment of dietary behaviors was observed in patients with concomitant MS. An unstable sense of self that can co-occur with schizophrenia may lead to derealization and depersonalization, resulting in distorted perception of one’s body and behavior [47]. A recommendation to introduce a healthy diet limited to a note advising a patient “not to eat sweets and increase physical activity” may not be a sufficient form of intervention. Given that a diet-related lifestyle change should go way beyond normalization of body weight, it should include improvement in eating attitudes and associated emotional regulation, reduction in food-related cognitive distortions, not to mention proper nutrition re-education. In our sample, schizophrenia patients without MS manifested better insight into their dietary behaviors. They were aware of their negative eating habits, which can be a starting point for introducing change. The use of different motivation techniques can enhance patient learning skills and attention. There is evidence of significant relationships between external motivation and cognitive improvement [23,48]. The dopaminergic system plays a key role in motivational processes, especially the black matter and the ventral tegmental area, which are responsible for predicting reward [49]. However, motivation is a representation of numerous processes aimed at maintaining action. It is a complex interaction between physiological and social processes [22], which include adequate nutrition education tailored to the needs of each individual patient. When considering the use of Motivational Dialogue in a psychiatric setting, Villaume et al. highlight its desirable effects on treatment adherence, and, by extension, also compliance with established dietary recommendations [50]. The desire to start treatment and introduce dietary changes is strictly dependent on individual illness-related experiences and their mental representations. Understanding patient’s beliefs concerning the diagnosis, the underlying causes of the disease and its possible consequences can be crucial in planning therapy. It seems that it is the cognitive representation of the disease that tends to guide patient’s willingness to follow recommendations, engage in healthy behaviors, or use effective methods of managing illness stress [51]. It should not be assumed that patients with schizophrenia are unable to control their eating behaviors. Early nutrition education may encourage desirable changes and improve their sense of agency. However, effective interventions must be tailored to individual patient needs, which involves understanding patient’s background and devoting time to conduct an in-depth nutritional interview. There is clear evidence of the effect of pharmacotherapy on the occurrence of metabolic syndrome in schizophrenia, and a myriad of methods to reduce the adverse effects of administered drugs. In contrast, there is not yet an established model of interventions designed to help patients change their eating habits, indicating that this aspect of lifestyle of persons with schizophrenia requires further investigation.

## 5. Conclusions

Performed data analyses suggest the following conclusions:There are clear differences in applied culinary techniques between schizophrenia patients with and without MS, but the two groups do not differ significantly in terms of other eating habits.Positive eating habits are associated with better cognitive functioning in schizophrenia patients without MS, although the causal relationship remains unclear.Healthier eating patterns, in line with basic principles of healthy nutrition, are associated with fewer positive symptoms of schizophrenia, although the direction of the correlation, yet again, remains unclear.Schizophrenia patients without MS are aware of their nutritional errors, which can facilitate nutrition education and healthy lifestyle, and ultimately improve their metabolic parameters.

## 6. Limitations

The project had several limitations that could have affected the research procedure and final results. One of the limitations was the lack of a control group that would enable comparisons of nutritional behaviors between healthy individuals and people with schizophrenia. The authors did not decide to introduce a control group because of the assumed objectives of the study. The majority of female participants stemmed from the fact that male schizophrenia patients who met inclusion criteria refused to participate in the study—a phenomenon that might be worth exploring in the future. Further studies should thus involve larger samples and increase the percentage of male participants.

This study was cross-sectional, correlative in nature, precluding determination of the cause-effect relationships, and making it unclear what the primary and secondary variables were. Patients may have experienced evaluation apprehension as they were interviewed about health-related problems and subjected to performance assessment. The quantitative nature of the administered tests together with the imposed time limits could cause discomfort associated with the fear of poor results. The need to report eating habits and their subjective assessment could be associated with the desire to present oneself in a favorable light. It would be worthwhile to conduct a prospective study to further investigate the eating habits of persons with schizophrenia.

## Figures and Tables

**Table 1 jcm-09-00537-t001:** Sociodemographic characteristics of the sample, N (%) or M (SD).

Variable	Entire Sample, *n* = 87	w/MS, *n* = 60	w/o MS, *n* = 27	X^2^/Z	df	*p*
Age	41.67 (12.87)	43.23 (12.81)	38.19 (12.53)	−1.676	-	0.094
BMI	31.77 (5.83)	33.59 (5.38)	27.74 (4.71)	−4.569	-	<0.001
Sex						
Female	55 (63.2%)	36 (60.0%)	19 (70.4%)	0.861	1	0.472
Male	32 (36.8%)	24 (40.0%)	8 (29.6%)			
Education				2.501	3	0.475
Elementary	6 (6.9%)	3 (5.0%)	3 (11.1%)			
Vocational	18 (20.7%)	14 (23.3%)	4 (14.8%)			
Secondary	39 (44.8%)	25 (41.7%)	14 (51.9%)			
Higher	24 (27.6%)	18 (30.0%)	6 (22.2%)			
Place of residence						
Country	14 (16.1%)	9 (15.0%)	5 (18.5%)	7.049	4	0.133
Small town	1 (1.1%)	0 (0.0%)	1 (3.7%)			
Medium-size town	7 (8.0%)	3 (5.0%)	4 (14.8%)			
Large city	5 (5.7%)	5 (8.3%)	0 (0.0%)			
Very large city	60 (69.0%)	43 (71.7%)	17 (63.0%)			
Employment status						
Student	21 (24.1%)	17 (28.3%)	4 (14.8%)	6.015	3	0.111
Employed	4 (4.6%)	1 (1.7%)	3 (11.1%)			
Unemployed	14 (16.1%)	8 (13.3%)	6 (22.2%)			
Pensioner/on health benefit	48 (55.2%)	34 (56.7%)	14 (51.9%)			
Marital status						
Single	40 (46.0%)	25 (41.7%)	15 (55.6%)	3.150	3	0.369
In a relationship	35 (40.2%)	25 (41.7%)	10 (37.0%)			
Divorced	7 (8.0%)	5 (8.3%)	2 (7.4%)			
Widowed	5 (5.7%)	5 (8.3%)	0 (0.0%)			

Source: own elaboration.

**Table 2 jcm-09-00537-t002:** Comparison of selected eating behaviors in patients with and without MS (SH).

	Entire Sample, *n* = 87		w/MS, *n* = 60		w/o MS, *n* = 27		Pearson’s Chi-Square Test		
	N	%	N	%	N	%	X^2^	df	p
Consumed food products									
Fish	53	60.9	39	65	14	51.9	1.352	1	0.342
Salt	36	41.4	21	35	15	55.6	3.244	1	0.100
Juices	51	58.6	32	53.3	19	70.4	2.228	1	0.163
Coca-Cola	40	46.0	31	51.7	9	33.3	2.520	1	0.163
Water	76	87.4	53	88.3	23	85.2	0.167	1	0.733
Tea	66	75.9	44	73.3	22	81.5	0.675	1	0.589
Fruits	80	92.0	56	93.3	24	88.9	0.497	1	0.672
Vegetables	79	90.8	54	90	25	92.6	0.150	1	0.700
Fast foods	34	39.1	23	38.3	11	40.7	0.045	1	0.832
Sweets	77	88.5	54	90	23	85.2	0.424	1	0.493
Groats	57	65.5	38	63.3	19	10.4	0.408	1	0.629
Pasta	63	72.4	46	76.7	17	63	1.750	1	0.204
Meat	70	80.5	48	80	22	81.5	0.026	1	0.873
Dairy	79	90.8	53	88.3	26	96.3	1.414	1	0.426
Stimulants									
Alcohol	5	5.8	4	6.7	1	3.7	0.302	1	0.959
Cigarettes	40	46.0	32	53.3	8	29.6	4.212	1	0.062
Strong coffee	31	35.6	25	41.7	6	22.2	3.070	1	0.095
Strong tea	18	20.7	18	30	0	0	0.461	1	0.794
Snacking and preparation of meals									
Snacking	9	10.4	5	8.3	4	14.8	0.843	1	0.450
Self-preparation of meals	17	19.5	9	15.0	8	29.6	2.535	1	0.145
Applied culinary techniques									
Boiling	56	64.4	34	56.7	22	81.5	4.999	1	0.031
Simmering	17	19.5	12	20.0	5	18.5	0.026	1	0.873
Frying	76	87.4	57	95.0	19	70.4	10.227	1	0.003
Baking	29	33.3	17	28.3	12	44.4	2.175	1	0.150

Source: own elaboration.

**Table 3 jcm-09-00537-t003:** Comparison of schizophrenia patients with and without MS.

Variable	w/MS, *n* = 60		w/o MS, *n* = 27		Mann-Whitney U Test	
	Min-Max	M (SD)	Min-Max	M (SD)	Z	p
1. Positive eating habits	3.00–11.00	7.28 (1.49)	3.00–11.00	7.74 (2.03)	−1.559	0.119
2. Negative eating habits	4.00–11.00	7.07 (1.52)	3.00–10.00	6.56 (1.85)	−0.943	0.346
3. Subjective assessment	1.00–5.00	3.02 (0.89)	1.00–5.00	3.26 (1.10)	−1.178	0.239
4. PANSS–positive	7.00–16.00	8.47 (2.0)	7.00–10.00	7.85 (1.17)	−1.236	0.217
5. PANSS–negative	6.00–28.00	12.08 (4.14)	7.00–20.00	11.48 (4.07)	−0.673	0.501
6. PANSS–general	13.00–35.00	21.40 (4.70)	2.00–−0.572	20.78 (5.55)	33.00	0.567
7. SCWT–time	0.00–71.52	23.22 (8.84)	17.81–35.79	22.66 (4.58)	- 0.071	0.944
8. SCWT–errors	0.00–3.00	0.13 (0.50)	0.00–0.00	0.00 (0.00)	−1.507	0.132
9. Digit span forward	3.00–13.00	5.93 (1.73)	4.00–9.00	5.69 (1.57)	−0.730	0.460
10. Digit span backward	2.00–9.00	4.40 (1.65)	2, 00–8.00	4.58 (1.60)	−0.542	0.588
11. Digit span total	6.00–22.00	10.30 (2.82)	4.00–17.00	10.12 (3.05)	−0.175	0.861
12. VFT–animals	3.0–27.0	14.83 (4.81)	5.00–21.00	14.73 (3.87)	−0.160	0.873
13. VFT—phonemic	0.0–17.0	7.75 (3.59)	3.00–15.00	7.35(3.17)	−0.656	0.512
14. VFT—sharp objects	1.0–14.0	6.35 (2.77)	2.00–−0.602	6.73 (2.74)	13.00	0.547

**Table 4 jcm-09-00537-t004:** Variable correlations in schizophrenia patients with MS (*N* = 60).

	1	2	3	4	5	6	7	8	9	10	11	12	13
1. Positive eating habits	-												
2. Negative eating habits	0.058	-											
3. Subjective assessment	0.087	0.070	-										
4. PANSS–positive	−0.058	−0.138	0.277 *	-									
5. PANSS–negative	0.113	−0.305 *	0.137	0.575 ***	-								
6. PANSS–general	0.088	−0.161	0.114	0.501 ***	0.603 ***	-							
7. SCWT–time	0.072	−0.059	0.229	0.049	0.058	0.027	-						
8. SCWT–errors	−0.053	0.052	0.064	0.199	0.055	0.066	0.041	-					
9. Digit span forward	−0.123	−0.045	−0.058	−0.075	−0.092	−0.044	−0.084	−0.134	-				
10. Digit span backward	−0.105	−0.076	−0.065	−0.215	−0.196	−0.258 *	−0.244	−0.114	0.30	-			
11. Digit span total	−0.104	−0.089	−0.106	−0.195	−0.192	−0.203	−0.194	−0.129	0.764 ***	0.829 ***	-		
12. VFT–animals	−0.021	0.132	−0.141	−0.354 **	−0.240	−0.330 *	−0.450 ***	−0.129	0.268 *	0.333 **	0.364 **	-	
13. VFT—phonemic	0.011	0.110	−0.124	−0.332 *	−0.327 *	−0.221	−0.332 *	−0.263 *	0.350 *	0.466 ***	0.518 ***	0.590 ***	-
14. VFT—sharp objects	0.107	0.171	−0.070	−0.257 *	−0.284 *	−0.271 *	−0.450 ***	0.093	0.280 *	0.425 **	0.415 **	0.608 ***	0.496 ***

* *p* < 0.05; ** *p* < 0.01; *** *p* < 0.001. Source: own elaboration.

**Table 5 jcm-09-00537-t005:** Variable correlations in schizophrenia patients without MS (*N* = 27).

	1	2	3	4	5	6	7	8	9	10	11	12	13
1. Positive eating habits	-												
2. Negative eating habits	0.255	-											
3. Subjective assessment	0.099	−0.485 *	-										
4. PANSS–positive	−0.501 **	−0.095	0.027	-									
5. PANSS–negative	−0.369	0.297	−0.306	0.619 **	-								
6. PANSS–general	−0.515 **	0.131	−0.262	0.663 ***	0.659 ***	-							
7. SCWT–time	−0.616 **	0.016	−0.211	0.234	0.371	0.220	-						
8. SCWT–errors	-	-	-	-	-	-	-	-	-	-		-	-
9. Digit span forward	−0.094	−0.354	−0.110	0.060	−0.020	0.104	0.073	-	-				
10. Digit span backward	0.015	−0.240	−0.147	−0.110	−0.150	−0.163	−0.012	-	0.550 **	-			
11. Digit span total	−0.048	−0.320	−0.162	−0.010	−0.143	−0.004	0.037	-	0.858 ***	0.867 ***	-		
12. VFT–animals	0.383	0.091	0.080	−0.424 *	−0.520 **	−0.256	−0.444 *	-	0.383	0.269	0.366	-	
13. VFT—phonemic	0.153	0.031	−0.154	−0.391 *	−0.245	−0.208	0.036	-	0.189	−0.011	−0.057	0.363	-
14. VFT—sharp objects	0.224	−0.112	0.195	−0.256	−0.320	0.009	−0.202	-	−0.167	−0.068	−0.116	0.547 **	0.300

* *p* < 0.05; ** *p* < 0.01; *** *p* < 0.001. Source: own elaboration.

## Data Availability

The data set generated and analyzed in this study is not publicly available due to participant data confidentiality.

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
