# Peer review of "Dietary Behaviors and Metabolic Syndrome in Schizophrenia Patients"

_jcm, 2020, doi:10.3390/jcm9020537_

Round 1
Reviewer 1 Report
The authors corrected the manuscript according to the most of my comments
Author Response
The authors thank you for all comments and positive consideration of previously introduced changes in the manuscript.
Reviewer 2 Report
In this study authors have investigated eating habits of Schizophrenia patient's with metabolic syndrome (MS). In this current study authors also studied effect of dietary habits on cognitive performance, symptom severity.
Authors observed no significant differences in dietary habits between investigated schizophrenia patients with and without MS. However, the authors observed a positive eating habits correlated with better cognitive performance. Next, authors observed that healthy eating habits were linked with the Stroop test which depicts processing and working memory. They observed also improved verbal fluency linked to healthier eating habits.
This manuscript is well written and presented in a nice way. I have few queries related to this study
1. People with schizophrenia show a high incidence of metabolic syndrome, associated with a high mortality from cardiovascular disease. Did authors find that any of the study subjects have cardiovascular disease and if they have, did they consider this or they simply excluded these samples.
2. Early life stress (ELS) also contribute to Schizophrenia. Did authors find patients with early life stress and corrected their data analysis for ELS. If they observed Schizophrenia patients with ELS then it would be interesting to investigate association of ELS, MS and dietary habit
3. Authors mentioned that they included 87 patients. However, in table 1 it is written entire sample =83. I assume it is a typing error.
Author Response
Point 1: People with schizophrenia show a high incidence of metabolic syndrome, associated with a high mortality from cardiovascular disease. Did authors find that any of the study subjects have cardiovascular disease and if they have, did they consider this or they simply excluded these samples.
Response 1:
Patients with serious heart disease were excluded from the study. Thank you for paying attention to the exclusion criteria. They were placed in the manuscript. “Exclusion criteria were other serious medical conditions that could require specialized dietary regimen or significantly affect cognitive performance. Such as acute symptoms of schizophrenia (increases positive and negative symptoms), eating disorders, known food intolerances, dementia, addiction to psychoactive substances, kidney and liver problems, thyroid dysfunction, syndromes of cardiac function disorders, oncological diseases” – lines number 133-138
Point 2: Early life stress (ELS) also contribute to Schizophrenia. Did authors find patients with early life stress and corrected their data analysis for ELS. If they observed Schizophrenia patients with ELS then it would be interesting to investigate association of ELS, MS and dietary habit
Response 2:
The authors have not studied the impact of ELS on eating behavior, but they thank you very much for the reviewer's suggestion and will take it into account when planning further studies.
Point 3: Authors mentioned that they included 87 patients. However, in table 1 it is written entire sample =83. I assume it is a typing error.
Response 3:
After checking the database and recalculating the sample, it turned out to be a typing error. The information in the manuscript has been corrected.

Reviewer 3 Report
The authors have appropriately responded to the reviewers' comments.
Author Response
The authors thank you for all comments and positive consideration of previously introduced changes in the manuscript.
This manuscript is a resubmission of an earlier submission. The following is a list of the peer review reports and author responses from that submission.
Round 1
Reviewer 1 Report
Eating habits of the patients with schizophrenia may influence their physical condition and somatic health. Moreover, patients cognitive performance may correlate with obesity and/or metabolic syndrom. This is an interesting study exploring the issue of dietary habits of patients with schizophrenia together with their cognitive functions, presence of metabolic syndrom and the clinical picture of the disease.
However, there are some points that need revision.
The use of control group of healthy persons could improve the study, the problem should at least be discussed as limitation Criteria of metabolic syndrom should be presented in more details Authors conclude that better eating habits correlate with better cognitive performance in patients without MS - the correlation is rather weak, so conclusions go too far, probably persons with better cognition have better dietary habits, it should be discussed in more details Similarly, the conclusion about correlation of positive dietary habits and positive symptoms should be discussed and the conclusions in general should be reworded Adding some data of patients biochemistry could be helpful, e.g. lipid panel, glucose level, at least BMI should be addedReviewer 2 Report
The study aimed to explore correlates of metabolic syndrome and negative eating habits in subjects with schizophrenia. The sample of 87 subjects was divided in those with (N=60) and without MS (N=27). The two groups were compared on symptoms, cognitive performance and dietary habits.
Apart from the limited sample size for this type of study and for the prevalence of females in the sample which limits the generalizability of results, the main concern with this study is the absence of comparison between the two groups on pharmacotherapy. The authors state that administered drugs were clozapine, olanzapine, quetiapine and aripiprazole which differ greatly in their propensity to cause obesity and dyslipidemia. This comparison should be added. Also the body weight or body mass index should be reported.
Finally, the correlations might be investigated in all subjects to avoid the reduced power due to the small sample size and restricted variance in homogeneous subgroups.
The study limitations (cross-sectional design, limited sample size and prevalence of females) should be mentioned in the discussion.
Minor points
Please correct typos and grammatical errors: e.g., “Patient is satisfied with their eating habits” should be “Patient is satisfied with his/her eating habits”.
The PANSS scale translated by Małgorzata Rzewuska was validated? A reference should be provided.